# FASTER-VPS: ACCELERATING OBJECT-LEVEL INTERPRETATION OF MULTIMODAL FOUNDATION MODELS

## ABSTRACT

Attribution is essential for interpreting object-level foundation models, yet existing methods struggle with the trade-off between efficiency and faithfulness. Gradient-based approaches are efficient but imprecise, while perturbation-based approaches achieve high fidelity at prohibitive cost. Visual Precision Search (VPS) represents the current state-of-the-art, but its greedy search requires a quadratic number of forward passes, severely limiting practicality. We introduce Faster-VPS, which replaces VPS's greedy search with a novel Phase-Window (PhaseWin) algorithm. PhaseWin combines phased pruning, windowed fine-grained selection, and adaptive control mechanisms to approximate greedy attribution with near-linear complexity. Theoretically, Faster-VPS retains approximation guarantees under monotonous submodular conditions. Empirically, it achieves over 95% of VPS's faithfulness using only 20% of the computational budget, and consistently outperforms all other attribution baselines on tasks such as object detection and visual grounding with Grounding DINO and Florence-2. Faster-VPS thus establishes a new state-of-the-art in efficient and faithful attribution.

## 1 INTRODUCTION

Understanding the decision-making process of large-scale foundation models (Dwivedi et al., 2023; Gao et al., 2024) is a fundamental challenge in artificial intelligence. Attribution methods (Montavon et al., 2017; Yamauchi et al., 2024), which aim to identify the input features most relevant to a model's output, are our primary tools for this endeavor. Effective attribution is not merely an academic exercise; it is critical for debugging models, diagnosing failures, uncovering hidden biases from training data, and ensuring that model behavior aligns with human values and safety constraints (Miller et al., 2019; Feng et al., 2021; Wilson et al., 2023; Stocco et al., 2022; Shu et al., 2024). For instance, in applications like autonomous driving, faithful attribution for object detection models is essential for building trustworthy systems (Liang et al., 2021; 2022a; Wei et al., 2019; Liang et al., 2022b; Liu et al., 2023).

Attribution methods are broadly classified into two paradigms: gradient-based (Zhao et al., 2024a; Yamauchi, 2024) and perturbation-based (Petsiuk et al., 2018; 2021). While gradient-based methods are computationally efficient, they often struggle with issues like artifact effects and multimodal interactions in gradient transfer (Selvaraju et al., 2020; Zhao et al., 2024a; Jiang et al., 2024), producing attribution maps that lack precision. In contrast, perturbation-based methods, which measure the model's response to systematically masking parts of the input, generally achieve much higher faithfulness. However, their superior performance is crippled by a steep computational cost (Novello et al., 2022; Jiang et al., 2023; Shapley, 1953), as the search for the most informative features often requires thousands of forward passes. The core research challenge, therefore, is to drastically reduce this computational overhead, making the high faithfulness of perturbation methods practical for real-world use.

The current state-of-the-art in faithful attribution is Visual Precision Search (VPS) Chen et al. (2025). VPS provides a complete workflow for perturbation-based attribution, formulating the problem as maximizing a submodular-like objective function (Edmonds, 1970; Chen et al., 2024b). Its core component is a greedy search over candidate regions, which ensures high faithfulness but also dominates the runtime (Fujishige, 2005). This quadratic complexity of VPS remains the principal barrier to its adoption in real-world, time-sensitive scenarios. The critical question we address is thus:

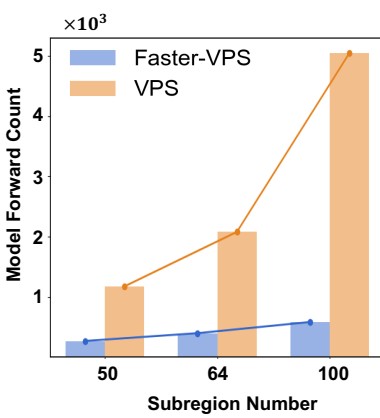 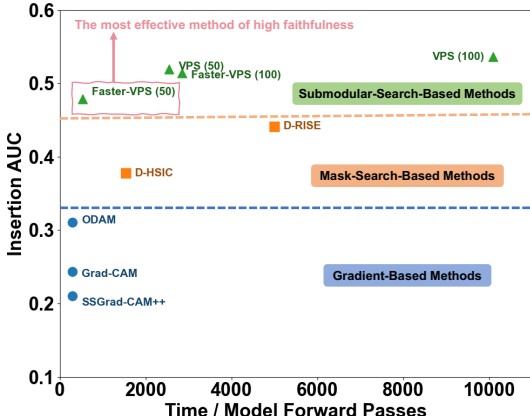

Figure 1: Efficiency–faithfulness trade-off in attribution methods. Left: comparison of model forward counts between VPS and Faster-VPS (window Size fixed as 16) across different subregion numbers. Right: comparison of Insertion AUC and computational cost among representative methods, where Faster-VPS achieves near-VPS faithfulness with a fraction of the computational budget.

*How can we retain the high faithfulness of a greedy perturbation search while drastically reducing its computational overhead?*

To overcome this limitation, we replace the greedy search with our Phase-Window (PhaseWin) algorithm, an efficient near-linear approximation to the greedy solution. PhaseWin begins with a phased coarse-to-fine search, where an anchor region is selected to set adaptive thresholds, pruning the majority of irrelevant candidates. The remaining high-potential regions are then processed by a windowed fine-grained selection, guided by two control mechanisms: a dynamic supervision policy that adaptively terminates phases with diminishing returns, and an annealed deferral strategy that helps escape poor local optima. By integrating PhaseWin into the VPS workflow, we assemble the complete Faster-VPS pipeline, which preserves the high faithfulness of VPS while drastically reducing its computational cost. As illustrated in Figure 1, VPS achieves excellent attribution quality but at the prohibitive cost of thousands of forward passes, whereas Faster-VPS reduces the overhead by an order of magnitude while retaining comparable faithfulness.

Our extensive experiments on object detection and visual grounding tasks with models such as Grounding DINO (Liu et al., 2024) and Florence-2 (Xiao et al., 2024) validate the effectiveness of Faster-VPS. Across MS COCO, LVIS, and RefCOCO, Faster-VPS achieves over 95% of VPS's faithfulness using only about 20% of the computational budget, establishing a new state-of-the-art in the efficiency–faithfulness trade-off. Moreover, ablation studies demonstrate that Faster-VPS can flexibly adjust speed–quality trade-offs: it can run in a highly accelerated mode for real-time use cases, or, when tuned for maximum quality, fully recover the original performance of VPS.

Our contributions are summarized as follows:

- **Faster-VPS pipeline.** We propose Faster-VPS, an accelerated variant of Visual Precision Search that reduces computational cost by an order of magnitude while preserving attribution faithfulness.

- **PhaseWin algorithm.** We introduce PhaseWin, a windowed search strategy with dynamic supervision and annealed deferral, which retains near-greedy optimality under submodular conditions.

- **Extensive validation.** Experiments on MS COCO, RefCOCO, and LVIS show that Faster-VPS attains over 95% of VPS's accuracy at only ∼20% cost, and can flexibly trade efficiency for precision.

## 2 RELATED WORK

**Object-level Foundation Models and Detection.** Object detection has evolved from two-stage Ren et al. (2016); He et al. (2018) and one-stage Redmon & Farhadi (2018); Tian et al. (2020) designs to Transformer-based architectures Carion et al. (2020). Multimodal pre-training Radford et al. (2021); Li et al. (2022); Wu et al. (2024a) has spurred object-level foundation models like Grounding DINO Liu et al. (2024) and Florence-2 Xiao et al. (2024), alongside unified decoders Zou et al. (2023a), large-scale models Wu et al. (2024b), and real-time open-vocabulary systems Cheng et al. (2024); Yao et al. (2024). The need for robustness and transparency in applications like contextual detection Zang et al. (2024), uncertainty-aware prediction Miller et al. (2019); Feng et al. (2021); Wilson et al.

(2023), and autonomous driving Wen et al. (2024); Chen et al. (2024a); Hu et al. (2023) highlights key challenges, as summarized in recent surveys Zou et al. (2023b); Liang et al. (2024).

**Explaining Object Detectors.** Explaining detector decisions is challenged by their intertwined localization and classification signals. Approaches range from adapting gradient-based attribution Gudovskiy et al. (2018); Selvaraju et al. (2020); Zhao et al. (2024a) and randomized perturbations Petsiuk et al. (2018; 2021) to refining Grad-CAM for spatial sensitivity Yamauchi & Ishikawa (2022); Yamauchi (2024); Chattopadhay et al. (2018). While some methods explore diverse rationales at high computational cost Jiang et al. (2023), recent state-of-the-art work uses causal search to generate high-fidelity explanations Chen et al. (2025). Other studies compare architectures Jiang et al. (2024), decompose representations Gandelsman et al. (2024), or analyze pixel collectives Yamauchi et al. (2024), with broader XAI surveys providing context Dwivedi et al. (2023); Gao et al. (2024).

**Submodular Function Maximization Algorithm.** Our research draws heavily on work that improves submodular function optimization (Edmonds, 1970; Fujishige, 2005). Since optimizing submodular functions doesn't necessarily mean optimizing AUC, this work can't be directly applied to attribution (Jegelka et al., 2011; Buchbinder et al., 2014). However, we considered how to exploit submodular properties (Wei et al., 2014; Breuer et al., 2020) and comprehensively designed our PhaseWin search algorithm, achieving a breakthrough in speed.

## 3 METHOD

### 3.1 PROBLEM FORMULATION

Given an input image $\mathbf{I} \in \mathbb{R}^{h \times w \times 3}$ and an object-level foundation model $f(\cdot)$, the detection result can be represented as $f(\mathbf{I}) = \{(b_i, c_i, s_i) \mid i = 1, 2, \ldots, N\}$, where $b_i$ denotes the bounding box, $c_i$ the predicted class label, and $s_i$ the confidence score of object $i$. We aim to explain the model's prediction for a specific target $(b_t, c_t, s_t)$ by selecting a sequence of critical input regions whose progressive insertion maximizes the model's confidence on the target.

We partition $\mathbf{I}$ into $m$ disjoint sub-regions $\mathcal{V} = \{\mathbf{I}_1^s, \ldots, \mathbf{I}_m^s\}$, and define an *ordered* subset $\mathcal{S} = (s_1, \ldots, s_k)$, where $s_i \in \mathcal{V}$. For a given ordering $\mathcal{S}$, let $F_\mathcal{S}(j) = f\left(\bigcup_{i=1}^{j} \mathbf{I}_{s_i}^s; b_t, c_t\right)$ denote the detection confidence after inserting the first $j$ regions in $\mathcal{S}$.

Our objective is to maximize the cumulative confidence along the insertion trajectory. Specifically, let $|\mathbf{I}_j^s|$ denote the pixel area of region $s_j$, and let $A = \sum_{r=1}^{m} |\mathbf{I}_r^s| = |\mathbf{I}|$ be the total image area. We define the optimal ordered subset $\mathcal{S}^*$ as:

$$\mathcal{S}^* = \arg \max_{\substack{\mathcal{S}=(s_1,\ldots,s_k) \\ \mathcal{S} \subseteq \mathcal{V}}} \sum_{j=1}^{k} \frac{|\mathbf{I}_{s_j}^s|}{A} F_\mathcal{S}(j),$$

where $F_\mathcal{S}(j)$ is the model confidence after inserting the first $j$ regions in $\mathcal{S}$.

This formulation explicitly treats the problem as an *ordered subset optimization*, where the evaluation depends on the insertion order.

### 3.2 SCORING FUNCTION

We adopt the scoring function $\mathcal{F}$ from VPS Chen et al. (2025), which is designed to identify critical regions for object detection. The function intelligently combines two complementary metrics: a **clue score** that quantifies how well a set of regions $S$ directly supports the target detection, and a **collaboration score** that measures the synergistic importance of $S$ by evaluating the performance drop when it is removed. Although $\mathcal{F}$ is not strictly submodular, its local submodularity makes it amenable to accelerated greedy search algorithms like our proposed PhaseWin method. The final objective function is:

$$\mathcal{F}(S, \mathbf{b}_{\text{target}}, c) = s_{\text{clue}}(S, \mathbf{b}_{\text{target}}, c) + s_{\text{colla}}(S, \mathbf{b}_{\text{target}}, c).$$

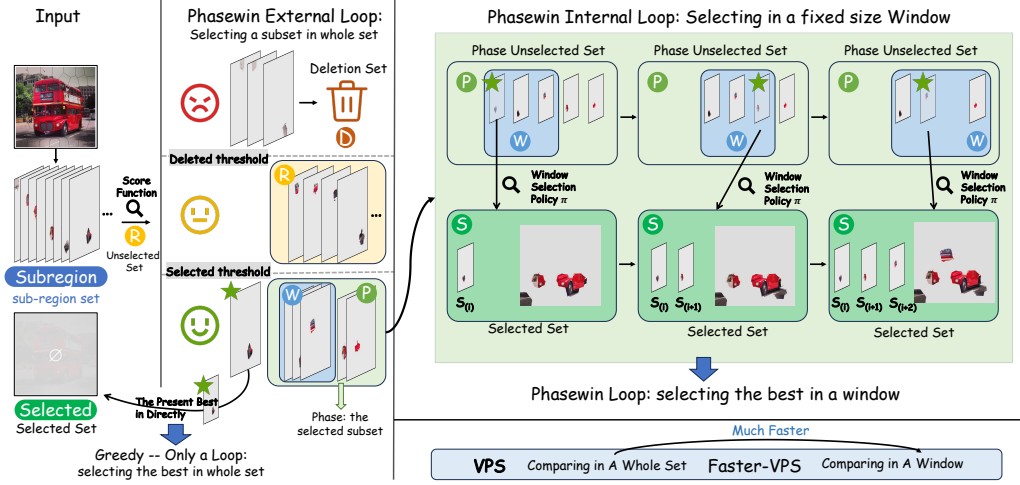

Figure 2: **PhaseWin Workflow.** The algorithm alternates between (i) selecting an *anchor* region to set adaptive thresholds, (ii) pruning uninformative regions, and (iii) applying a windowed fine-grained selection with dynamic supervision.

### 3.3 PHASE-WINDOW ACCELERATED SEARCH

For maximizing the ordered insertion-AUC objective, a naive greedy search that evaluates all remaining candidates at each step is theoretically optimal, but its $\mathcal{O}(k \cdot m)$ scoring cost is prohibitive in practice. We propose the **Phase-Window (PhaseWin) Search**, an efficient approximation that matches greedy performance while reducing the number of expensive scoring function calls by an order of magnitude.

PhaseWin's acceleration stems from a phased, coarse-to-fine search strategy, illustrated in Figure 2. The algorithm operates in phases, each beginning with a full evaluation to find a high-confidence *anchor* region. The gain from this anchor is used to set adaptive thresholds that aggressively prune the search space, creating a compact, high-potential candidate pool for the next stage. This high-level process is detailed in Algorithm 1. The core of our method lies in the `WindowSelection`

---

**Algorithm 1** PhaseWin: Phase-Window Accelerated Search

**Require:** Candidate set $\mathcal{V}$, target size $k$, scoring function $\mathcal{F}(\cdot)$
**Ensure:** Ordered subset $S$
1: $S \leftarrow \emptyset; \quad D \leftarrow \emptyset; \quad \mathcal{R} \leftarrow \mathcal{V}; \quad \Delta_{\text{prev}} \leftarrow \infty$
2: **while** $|S| < k$ **and** $\mathcal{R} \neq \emptyset$ **do**
3: $\quad g_r \leftarrow \mathcal{F}(S \cup \{r\}) - \mathcal{F}(S)$ for all $r \in \mathcal{R}$
4: $\quad \alpha^\star \leftarrow \arg\max_{r \in \mathcal{R}} g_r; \quad S \leftarrow S \cup \{\alpha^\star\}; \quad \Delta_{\text{prev}} \leftarrow \max g_r$
5: $\quad \mathcal{R} \leftarrow \mathcal{R} \setminus \{\alpha^\star\}$
6: $\quad \tau_{\text{sel}}, \tau_{\text{del}} \leftarrow \text{AdaptiveThresholds}(\Delta_{\text{prev}})$
7: $\quad \mathcal{P}_t, \mathcal{D}_{\text{phase}} \leftarrow \text{PartitionCandidates}(\mathcal{R}, \tau_{\text{sel}}, \tau_{\text{del}})$
8: $\quad D \leftarrow D \cup \mathcal{D}_{\text{phase}}$
9: $\quad S_{\text{phase}} \leftarrow \text{WindowSelection}(\mathcal{P}_t, S, k, \mathcal{F}, \Delta_{\text{prev}})$
10: $\quad S \leftarrow S \cup S_{\text{phase}}$
11: $\quad \mathcal{R} \leftarrow \mathcal{R} \setminus S_{\text{phase}}$ ▷ Unselected candidates form the pool for the next phase
12: **end while**
13: **return** $S$

---

subroutine, which performs a fine-grained search on a pruned candidate pool. We begin with a sliding window $W$ containing top-ranked candidates, while the rest remain in a queue. A window policy $\pi(\cdot)$ is then applied to select a subset $A$; in practice, we mainly use two policies: (1) $\pi_{LG}$, which picks the top candidate, and (2) $\pi_{BA}$, which selects all candidates above an adaptive cut-off based on the window's maximum gain.

For each candidate $\alpha \in A$, we compute its true marginal gain $\Delta_i$ and evaluate it with two control mechanisms. First, the **stage-exit** rule halts the phase if $\Delta_i < \theta \cdot \Delta_{\text{ref}}$, preventing unnecessary computation on diminishing returns. Otherwise, the candidate is further checked by an **annealing delay**, which decides whether to accept it immediately or defer acceptance. Accepted candidates are added to the current solution, their gains update the reference value, and the window is replenished from the queue until the process completes.

## 3.4 THEORY ANALYSIS

Greedy search is both a curse and a shackle in the development of submodular function maximization algorithms: it has long been proven to be the optimal and fastest method to achieve the best possible approximation under polynomial-time constraints. We first restate the classic result as follows.

**Proposition 3.1.** *For maximizing a monotone submodular objective $\mathcal{F} : 2^{\mathcal{V}} \to \mathbb{R}_+$ under a cardinality constraint $k$, let $S_{\text{greedy}}$ denote the solution returned by the standard greedy algorithm and $S_{\text{OPT}}$ denote the optimal subset of size $k$. Then the greedy algorithm achieves the optimal approximation ratio:*

$$\mathcal{F}(S_{\text{greedy}}) \geq \left(1 - \frac{1}{e}\right)\mathcal{F}(S_{\text{OPT}}),$$

*and no polynomial-time algorithm can surpass this bound unless $P = NP$ (Nemhauser et al., 1978; Fujishige, 2005).*

Therefore, greedy selection serves as the *de facto* gold standard, and our analysis focuses on matching its empirical behavior while achieving substantial acceleration. Our phase-window accelerated search (PHASEWIN) is analogous to quicksort for sorting: it is extremely fast in typical cases, yet it still offers explicit approximation guarantees with the phase-supervised early exit mechanism enabled.

**Theorem 3.1** (Approximation Guarantee). *Let $S_{\text{PhaseWin}}$ denote the solution returned by PHASEWIN, and let $\theta \in [0, 1)$ be an upper bound on the fraction of phases where early exits occur due to dynamic supervision. If the objective $\mathcal{F}$ is monotone submodular, then*

$$\mathcal{F}(S_{\text{PhaseWin}}) \geq \left(1 - \frac{1}{e} - \mathbf{o}(1)\right)\mathcal{F}(S_{\text{OPT}}).$$

*Remark* 3.1. This $\mathbf{o}(1)$ quantity is actually determined by $\tau_{sel}, \tau_{del}, k, \theta$. We have put the proof of this theorem in Appendix E.

Table 1: **Comparison of approximation guarantee, complexity, and empirical acceleration.** $k$ denotes the subset size, $m$ the total candidate set size, and $\varepsilon$ the maximal early-exit ratio.

| Method | Approx. Guarantee | # Marginal Evals | Complexity | Empirical Speedup |
|---|---|---|---|---|
| Greedy | $(1 - 1/e)$ | $\mathcal{O}(mk)$ | Quadratic | $1\times$ |
| Lazy Greedy | $(1 - 1/e)$ | $\sim 0.7\,mk$ | Sub-quadratic | $\sim 1.3\times$ |
| **PhaseWin** | $(1 - 1/e - \varepsilon)$ | $\mathcal{O}(m)$ | Near-linear | **5–10×** |

**Time Complexity Analysis.** Since the forward evaluation of the scoring function $\mathcal{F}(\cdot)$ dominates runtime, we analyze complexity in terms of the number of calls to $\mathcal{F}$.

With dynamic supervision, each phase aggressively prunes the candidate pool and probabilistically terminates when marginal gains diminish. Let $N_{\text{exit}}$ be the expected number of early-exited phases. The expected number of calls is:

$$\mathbb{E}[\#\text{calls}] = O\left(m \cdot f(\omega) + m \cdot N_{\text{exit}}\right),$$

where $w$ is the window size, and f decided by $\pi$ we select. For $\pi_{\text{LG}}$, $f(\omega) = \omega$, for $\pi_{\text{BA}}$, $f(\omega) = log(\omega)$, so the effective complexity approaches $\mathcal{O}(m)$ if $\omega << m$.

Thus, PHASEWIN achieves *greedy-level accuracy* while reducing the number of expensive scoring calls by up to an order of magnitude in practice. The above theoretical analysis takes into account ideal situations and makes full use of the submodularity assumption. Our experiments confirm its high efficiency. The definitions of submodularity and supermodularity and their corresponding AUC curve properties are in Appendix F.

## 4 EXPERIMENTS

### 4.1 EXPERIMENTAL SETUP

We conduct a comprehensive evaluation of our method on object detection and referring expression comprehension (REC) tasks. The experiments are performed using two powerful object-level foundation models: Grounding DINO (Liu et al., 2024) and Florence-2 (Xiao et al., 2024).

Table 2: Comparison on three datasets for correctly detected or grounded samples using Grounding DINO.

| Datasets | Methods | Faithfulness Metrics | | | | | | | Location Metrics | | Efficiency Metrics | |
|---|---|---|---|---|---|---|---|---|---|---|---|---|
| | | Ins. (↑) | Del. (↓) | Ins. (class) (↑) | Del. (class) (↓) | Ins. (IoU) (↑) | Del. (IoU) (↓) | Ave. high. score (↑) | Point Game (↑) | Energy PG (↑) | MEC_ave (↓) | A-C ratio (↑) |
| MS COCO (Lin et al., 2014) (Detection task) | Grad-CAM (Selvaraju et al., 2020) | 0.2436 | 0.1526 | 0.3064 | 0.2006 | 0.6229 | 0.5324 | 0.5904 | 0.1746 | 0.1463 | — | — |
| | SSGrad-CAM++ (Yamauchi & Ishikawa, 2022) | 0.2107 | 0.1778 | 0.2639 | 0.2314 | 0.5981 | 0.5511 | 0.5886 | 0.1905 | 0.1293 | — | — |
| | D-RISE (Petsiuk et al., 2018) | 0.4412 | 0.0402 | 0.5081 | 0.0886 | 0.8396 | 0.3642 | 0.6215 | 0.9497 | 0.1850 | 5000 | 0.88 |
| | D-HSIC (Novello et al., 2022) | 0.3776 | 0.0439 | 0.4382 | 0.0903 | 0.8301 | 0.3301 | 0.5862 | 0.7328 | 0.1861 | 1536 | 2.46 |
| | ODAM (Zhao et al., 2024b) | 0.3103 | 0.0519 | 0.3655 | 0.0894 | 0.7869 | 0.3984 | 0.5865 | 0.5431 | 0.2034 | — | — |
| | VPS (50) Chen et al. (2025) | 0.5195 | **0.0375** | 0.5941 | **0.0835** | 0.8480 | **0.3044** | 0.6591 | 0.9841 | **0.2046** | 2548.8 | 2.04 |
| | Faster-VPS (50) | 0.4785 | 0.0424 | 0.5562 | 0.0898 | 0.8323 | 0.3116 | 0.6353 | **0.9894** | 0.1843 | 536.8 | 8.92 |
| | VPS (100) (Chen et al., 2025) | **0.5459** | **0.0375** | **0.6204** | 0.0882 | **0.8581** | 0.3300 | **0.6873** | **0.9894** | **0.2046** | 10100 | 0.54 |
| | Faster-VPS (100) | 0.5141 | 0.0410 | 0.5890 | 0.0907 | 0.8505 | 0.3400 | 0.6644 | **0.9894** | 0.1628 | 2853.4 | 1.81 |
| RefCOCO (Kazemzadeh et al., 2014) (REC task) | Grad-CAM (Selvaraju et al., 2020) | 0.3749 | 0.4237 | 0.4658 | 0.5194 | 0.7516 | 0.7685 | 0.7481 | 0.2380 | 0.2171 | — | — |
| | SSGrad-CAM++ (Yamauchi & Ishikawa, 2022) | 0.4113 | 0.3925 | 0.5008 | 0.4851 | 0.7700 | 0.7588 | 0.7561 | 0.2820 | 0.2262 | — | — |
| | D-RISE (Petsiuk et al., 2018) | 0.6178 | 0.1605 | 0.7033 | 0.3396 | 0.8606 | 0.5164 | 0.8471 | 0.9400 | 0.2870 | 5000 | 1.24 |
| | D-HSIC (Novello et al., 2022) | 0.5491 | 0.1846 | 0.6295 | 0.3509 | 0.8504 | 0.5120 | 0.7739 | 0.7900 | 0.3190 | 1536 | 3.57 |
| | ODAM (Zhao et al., 2024b) | 0.4778 | 0.2718 | 0.5620 | 0.3757 | 0.8217 | 0.6641 | 0.7425 | 0.6320 | 0.3529 | — | — |
| | VPS (50) (Chen et al., 2025) | 0.7278 | **0.1240** | 0.7995 | 0.2473 | 0.8961 | **0.5053** | 0.8770 | **0.9580** | 0.3738 | 2290.6 | 3.18 |
| | Faster-VPS (50) | 0.7013 | 0.1473 | 0.7794 | 0.2747 | 0.8862 | 0.5273 | 0.8654 | **0.9580** | 0.3530 | 630.1 | 11.13 |
| | VPS (100) (Chen et al., 2025) | **0.7419** | 0.1250 | **0.8080** | **0.2457** | 0.9050 | 0.5103 | **0.8842** | 0.9460 | 0.3566 | 10100 | 0.73 |
| | Faster-VPS (100) | 0.7377 | 0.1529 | 0.8046 | 0.2823 | **0.9054** | 0.5466 | 0.8813 | 0.9360 | 0.3076 | 3382.5 | 2.18 |
| LVIS V1 (Gupta et al., 2019) (rare) (Zero-shot det. task) | Grad-CAM (Selvaraju et al., 2020) | 0.1253 | 0.1294 | 0.1801 | 0.1814 | 0.5657 | 0.5910 | 0.3549 | 0.1151 | 0.0941 | — | — |
| | SSGrad-CAM++ (Yamauchi & Ishikawa, 2022) | 0.1253 | 0.1254 | 0.1765 | 0.1775 | 0.5800 | 0.5691 | 0.3504 | 0.1091 | 0.0931 | — | — |
| | D-RISE (Petsiuk et al., 2018) | 0.2808 | 0.0289 | 0.3348 | 0.0835 | 0.8303 | 0.3174 | 0.4289 | 0.9697 | 0.1462 | 5000 | 0.56 |
| | D-HSIC (Novello et al., 2022) | 0.2417 | 0.0353 | 0.2912 | 0.0928 | 0.8187 | 0.3550 | 0.4044 | 0.8303 | 0.1730 | 1536 | 1.57 |
| | ODAM (Zhao et al., 2024b) | 0.2009 | 0.0410 | 0.2478 | 0.0844 | 0.7770 | 0.4082 | 0.3694 | 0.6061 | **0.2050** | — | — |
| | VPS (50) (Chen et al., 2025) | 0.3411 | **0.0265** | 0.3995 | 0.0805 | 0.8372 | 0.2986 | 0.4654 | **0.9939** | 0.1439 | 2544.6 | 1.34 |
| | Faster-VPS (50) | 0.3071 | 0.0303 | 0.3645 | 0.0893 | 0.8245 | 0.3097 | 0.4325 | **0.9939** | 0.1369 | 465.9 | 6.59 |
| | VPS (100) (Chen et al., 2025) | **0.3695** | 0.0277 | **0.4275** | **0.0799** | 0.8479 | 0.3242 | **0.4969** | 0.9758 | 0.1785 | 10100 | 0.37 |
| | Faster-VPS (100) | 0.3363 | 0.0309 | 0.3944 | 0.0839 | 0.8379 | 0.3374 | 0.4688 | 0.9697 | 0.1175 | 2726.8 | 1.23 |

**Datasets and Baselines.** We conduct experiments on three benchmarks. MS COCO 2017 (Lin et al., 2014) covers 80 object classes; we sample correctly detected, misclassified, and undetected instances for evaluation. LVIS V1 (Gupta et al., 2019) spans 1,203 categories with 337 rare ones, where Grounding DINO (Liu et al., 2024) is used for zero-shot detection. RefCOCO (Kazemzadeh et al., 2014) is adopted for the REC task, including both correct and incorrect grounding cases. We compare against representative attribution methods: gradient-based (Grad-CAM (Selvaraju et al., 2020), SSGrad-CAM++ (Yamauchi & Ishikawa, 2022), ODAM (Zhao et al., 2024b)), perturbation-based (D-RISE (Petsiuk et al., 2018), D-HSIC (Novello et al., 2022)), and the original Visual Precision Search (VPS (Chen et al., 2025)), a greedy quadratic algorithm that serves as our acceleration target.

**Implementation Details.** For Faster-VPS, we adopt a default window size of 16 for 50 sub-regions and 32 for 100 sub-regions. Since the score function is not strictly monotonic submodular, we implement the stopping criterion using a numerically stable ratio-based formulation: $\frac{S_{k-2}}{S_{k-1}} - \frac{S_{k-1}}{S_k} \leq \tau$, where we set $\tau = 0.025$ for 50 sub-regions and $\tau = 0.01$ for 100 sub-regions. Complete implementation details are provided in Appendix D.

## 4.2 EVALUATION METRICS

We evaluate the quality of attributions along three key axes: faithfulness, localization accuracy, and computational efficiency. This enables a holistic comparison of Faster-VPS against all baselines.

**1.Faithfulness.** We adopt standard insertion and deletion metrics to evaluate how well attribution maps reflect the model's reasoning, applied to both classification confidence and IoU. We also report the highest box confidence (IoU > 0.5) and the Explaining Successful Rate (ESR) for failure cases.
**2.Localization Accuracy.** We follow prior work and use the Point Game (Zhang et al., 2018) and Energy Point Game metrics (Wang et al., 2020) to quantify the alignment between attribution maps and ground-truth objects. **3.Efficiency.** We measure runtime efficiency using Model Evaluation Count (MEC), where one unit corresponds to a single forward pass. To combine accuracy and cost, we also report the Accuracy–Cost Ratio (AC-Ratio). Details of the above metrics are provided in Appendix C.

## 4.3 FAITHFULNESS ANALYSIS ON CORRECT SAMPLES

We follow the experimental design of VPS and conduct faithfulness, locality, and efficiency tests on correct detection, and faithfulness, error correction, and efficiency tests on misclassification and non-detection cases on three datasets.

### 4.3.1 CORRECT INTERPRETATION ON GROUNDING DINO

We follow the experimental design of VPS and conduct faithfulness, locality, and efficiency tests on correct detection, and faithfulness, error correction, and efficiency tests on misclassification

and non-detection cases on three datasets. Table 2 summarizes the results on correctly detected or grounded samples across three benchmarks. On the MS COCO detection task, Faster-VPS substantially improves efficiency while maintaining comparable faithfulness. Under the 50-region setting, it reduces the average model evaluations from 2548.8 to 536.8, a 4.7× reduction, with only a minor decrease in the Insertion score (0.5195 to 0.4785). This trade-off yields a marked improvement in the A-C ratio from 2.04 to 8.92. For the RefCOCO referring expression comprehension benchmark, Faster-VPS achieves similar faithfulness to VPS, with an Insertion score of 0.7013 versus 0.7278, while reducing model evaluations from 2290.6 to 630.1. This efficiency gain elevates the A-C ratio from 3.18 to 11.13, showing that Faster-VPS produces high-quality attributions at a fraction of the cost. On the challenging LVIS v1 rare-class detection task, both VPS and Faster-VPS show reduced overall faithfulness due to long-tail distributions. Nevertheless, Faster-VPS lowers the computation demand from 2544.6 to 465.9 evaluations in the 50-region setting, improving the A-C ratio from 1.34 to 6.59. These results highlight that the efficiency benefits of Faster-VPS become particularly valuable in computationally intensive scenarios, making attribution analysis more practical at scale.

### 4.3.2 CORRECT INTERPRETATION ON FLORENCE-2

Table 3 reports results on MS COCO and Ref-COCO when using Florence-2 as the backbone. Across both datasets, Faster-VPS achieves faithfulness scores that are highly comparable to VPS. On MS COCO, Faster-VPS attains an Insertion score of 0.7615 versus 0.7678 from VPS, with a slightly lower Deletion value (0.0474 vs. 0.0550). Similarly, on RefCOCO, Faster-VPS produces an Insertion of 0.8312 against 0.8301 from VPS, with a minor increase in Deletion. These results indicate that the acceleration strategy preserves the fidelity of VPS

Table 3: Evaluation of faithfulness (Insertion/Deletion AUC) and efficiency metrics on MS COCO and RefCOCO validation sets (Florence-2).

| Datasets | Methods | Faithfulness Metrics | | Efficiency Metrics | |
|---|---|---|---|---|---|
| | | Insertion (↑) | Deletion (↓) | MEC$_{ave}$ (↓) | A-C ratio (↑) |
| MS COCO (Detection task) | D-RISE | 0.7477 | 0.0972 | 5000 | 1.50 |
| | D-HSIC | 0.5345 | 0.2730 | **1536** | 3.48 |
| | VPS (50) | **0.7678** | 0.0550 | 2548.1 | 2.98 |
| | Faster-VPS (50) | 0.7615 | **0.0474** | 2184.1 | **3.49** |
| RefCOCO (REC task) | D-RISE | 0.7922 | 0.3505 | 5000 | 1.24 |
| | D-HSIC | 0.7639 | 0.3560 | **1536** | **3.57** |
| | VPS (50) | 0.8301 | **0.1159** | 2547.8 | 3.25 |
| | Faster-VPS (50) | **0.8312** | 0.1205 | 2349.1 | 3.53 |

almost entirely. When contrasted with perturbation-based baselines, Faster-VPS consistently delivers higher faithfulness while requiring fewer model evaluations than D-RISE, and achieves efficiency comparable to D-HSIC but with stronger interpretability. The A-C ratio also reflects this balance: Faster-VPS improves upon VPS (3.49 vs. 2.98 on COCO; 3.53 vs. 3.25 on RefCOCO), showing more favorable faithfulness-to-cost trade-offs. It is worth noting that the acceleration gains are less pronounced compared to Grounding DINO. Florence-2 exhibits behavior that is nearly globally supermodular, while our acceleration relies on exploiting local submodularity. As discussed in appendix F, this structural property limits the extent of achievable speedup. Nevertheless, Faster-VPS remains a strong alternative to VPS, offering similar interpretability at reduced computational cost and outperforming other baselines across both benchmarks.

## 4.4 FAILURES INTERPRETATION

### 4.4.1 REC FAILURES INTERPRETATION

Table 4 presents results on RefCOCO samples where Grounding DINO produces incorrect grounding. Compared with gradient-based baselines such as Grad-CAM and ODAM, both VPS and Faster-VPS yield substantially higher insertion scores and average confidence, indicating that search-based attribution is better suited for recovering meaningful evidence under failure cases. Perturbation-based

Table 4: RefCOCO (REC task): Faithfulness metrics and efficiency (Grounding DINO).

| Datasets | Methods | Faithfulness Metrics | | | Efficiency Metrics | |
|---|---|---|---|---|---|---|
| | | Ins. (↑) | Ins. (class) (↑) | Ave. high.score (↑) | MEC$_{ave}$ (↓) | A-C ratio (↑) |
| RefCOCO (REC task) | Grad-CAM | 0.1536 | 0.2794 | 0.3295 | — | — |
| | SSGrad-CAM++ | 0.1590 | 0.2837 | 0.3266 | — | — |
| | D-RISE | 0.3486 | 0.4787 | 0.6096 | 5000 | 1.21 |
| | D-HSIC | 0.2274 | 0.3488 | 0.4495 | 1536 | 2.92 |
| | ODAM | 0.1793 | 0.3001 | 0.3453 | — | — |
| | VPS (100) | 0.4981 | 0.5990 | 0.7007 | 10100 | 0.69 |
| | Faster-VPS (50) | 0.4455 | 0.5537 | 0.6437 | 614.4 | **10.48** |
| | Faster-VPS (100) | **0.5047** | **0.6023** | **0.7116** | 3164.4 | 2.25 |

approaches like D-RISE and D-HSIC achieve moderate improvements, but remain less faithful overall. Between the two search variants, Faster-VPS attains attribution quality that is highly comparable to VPS. Under the 100-region setting, Faster-VPS slightly surpasses VPS in insertion and classification-based scores (0.5047 vs. 0.4981 and 0.6023 vs. 0.5990), while using fewer model evaluations (3164.4 vs. 10100). In the 50-region setting, Faster-VPS achieves somewhat lower insertion metrics than VPS but with drastically reduced computational demand (614.4 vs. 10100 evaluations). This efficiency translates into a much higher A-C ratio, rising from 0.69 with VPS to 10.48 with Faster-VPS. These

results suggest that Faster-VPS can provide nearly the same level of interpretability as the greedy search, while significantly reducing the computational cost. This advantage is especially valuable when analyzing mis-grounded instances, where large-scale evaluation would otherwise be prohibitive.

### 4.4.2 MISCLASSIFIED DETECTION FAILURES INTERPRETATION

Table 5 reports results on misclassified samples from MS COCO and LVIS. Gradient-based methods such as Grad-CAM and ODAM show limited utility in this setting, while perturbation-based baselines (D-RISE and D-HSIC) provide moderate improvements in insertion and class-specific scores. Both VPS and Faster-VPS yield higher overall faithfulness, indicating that search-based approaches are better suited to reveal discriminative regions responsible for misclassification. On

Table 5: MS COCO and LVIS (misclassified samples): Faithfulness metrics and efficiency (Grounding DINO)

| Datasets | Methods | Faithfulness Metrics | | | | Efficiency Metrics | |
|---|---|---|---|---|---|---|---|
| | | Ins. (↑) | Ins. (class) (↑) | Ave. high. score (↑) | ESR (↑) | MEC$_{ave}$ (↓) | A-C ratio (↑) |
| MS COCO (Detection task) | Grad-CAM | 0.1091 | 0.1478 | 0.3102 | 38.38% | — | — |
| | SSGrad-CAM++ | 0.0960 | 0.1336 | 0.2952 | 33.51% | — | — |
| | D-RISE | 0.2170 | 0.2661 | 0.3603 | 50.26% | 5000 | 0.72 |
| | D-HSIC | 0.1771 | 0.2161 | 0.3143 | 34.59% | 1536 | 2.04 |
| | ODAM | 0.1129 | 0.1486 | 0.2869 | 32.97% | — | — |
| | VPS (100) | **0.3357** | **0.3967** | **0.4591** | **69.73%** | 10100 | 0.45 |
| | Faster-VPS (50) | 0.2614 | 0.3198 | 0.3770 | 51.35% | **477.3** | **7.90** |
| | Faster-VPS (100) | 0.3018 | 0.3583 | 0.4289 | 63.78% | 2595.0 | 1.65 |
| LVIS V1 (rare) (Zero-shot det. task) | Grad-CAM | 0.0503 | 0.0891 | 0.1564 | 12.50% | — | — |
| | SSGrad-CAM++ | 0.0574 | 0.0946 | 0.1580 | 11.84% | — | — |
| | D-RISE | 0.1245 | 0.1647 | 0.2088 | 28.95% | 5000 | 0.41 |
| | D-HSIC | 0.0963 | 0.1247 | 0.1748 | 16.45% | 1536 | 1.14 |
| | ODAM | 0.0575 | 0.0954 | 0.1520 | 9.21% | — | — |
| | VPS (100) | **0.1776** | **0.2190** | **0.2606** | **43.29%** | 10100 | 0.26 |
| | Faster-VPS (50) | 0.1394 | 0.1817 | 0.2119 | 36.63% | **426.5** | **5.20** |
| | Faster-VPS (100) | 0.1475 | 0.1845 | 0.2296 | 39.47% | 2204.8 | 1.04 |

MS COCO, VPS achieves the strongest raw faithfulness metrics, with an Insertion score of 0.3357 and an ESR of 69.73%. Faster-VPS produces slightly lower attribution quality under both 50- and 100-region settings, but substantially reduces computational requirements. In particular, Faster-VPS (50) lowers the average model evaluations from 10100 to only 477.3, raising the A-C ratio from 0.45 to 7.90. This demonstrates that Faster-VPS can provide competitive interpretability while making failure case analysis far more efficient. On LVIS rare-class detection, all methods perform worse due to the long-tail distribution, but the same trend holds. VPS delivers the highest insertion and ESR values, while Faster-VPS achieves comparable results at a fraction of the computational cost. Faster-VPS (50) reduces the model evaluations by over 20× compared to VPS, yielding an A-C ratio of 5.20 versus 0.26. These results show that Faster-VPS remains practical for large-scale misclassification analysis, where the quadratic cost of full VPS would be prohibitive.

### 4.4.3 UNDETECTED DETECTION FAILURES INTERPRETATION

Table 6 presents results on MS COCO and LVIS samples where the target objects are not detected. In this challenging setting, gradient-based baselines such as Grad-CAM and ODAM yield low insertion and class-specific scores, reflecting limited explanatory power. Perturbation-based methods (D-RISE and D-HSIC) offer some improvements, but remain costly or less stable. In contrast, both VPS and Faster-VPS deliver more reliable attribution maps, bet-

Table 6: MS COCO and LVIS (undetected failure samples): Faithfulness metrics and efficiency (Grounding DINO).

| Datasets | Methods | Faithfulness Metrics | | | | Efficiency Metrics | |
|---|---|---|---|---|---|---|---|
| | | Ins. (↑) | Ins. (class) (↑) | Ave. high. score (↑) | ESR (↑) | MEC$_{ave}$ (↓) | A-C ratio (↑) |
| MS COCO (Detection task) | Grad-CAM | 0.0760 | 0.1321 | 0.2153 | 16.44% | — | — |
| | SSGrad-CAM++ | 0.0671 | 0.1151 | 0.2124 | 16.44% | — | — |
| | D-RISE | 0.1538 | 0.2260 | 0.2564 | 26.94% | 5000 | 0.31 |
| | D-HSIC | 0.1101 | 0.1716 | 0.1945 | 13.56% | 1536 | 1.43 |
| | ODAM | 0.0745 | 0.1350 | 0.2037 | 13.78% | — | — |
| | VPS (100) | 0.2102 | 0.3011 | 0.3014 | 41.33% | 10100 | 0.21 |
| | Faster-VPS (50) | 0.1801 | 0.2641 | 0.2726 | 33.78% | **427.8** | **6.37** |
| | Faster-VPS (100) | **0.2156** | **0.3045** | **0.3289** | **44.44%** | 2160.2 | 1.52 |
| LVIS V1 (rare) (Zero-shot det. task) | Grad-CAM | 0.0291 | 0.0689 | 0.0901 | 5.43% | — | — |
| | SSGrad-CAM++ | 0.0292 | 0.0680 | 0.0897 | 5.24% | — | — |
| | D-RISE | 0.0703 | 0.1184 | 0.1312 | 18.73% | 5000 | 0.26 |
| | D-HSIC | 0.0516 | 0.0920 | 0.1168 | 13.48% | 1536 | 0.76 |
| | ODAM | 0.0283 | 0.0716 | 0.0851 | 4.68% | — | — |
| | VPS (100) | **0.1155** | **0.1886** | **0.1784** | **30.15%** | 10100 | 0.18 |
| | Faster-VPS (50) | 0.0787 | 0.1286 | 0.1309 | 17.04% | **348.4** | **3.76** |
| | Faster-VPS (100) | 0.0942 | 0.0069 | 0.1552 | 24.72% | 1509.1 | 1.03 |

ter capturing the evidence that is missing in undetected cases. On MS COCO, Faster-VPS achieves faithfulness that is close to VPS, with Insertion and ESR values of 0.2156 and 44.44% under the 100-region setting, slightly exceeding VPS. More importantly, it reduces the average model evaluations from 10100 to 2160.2, raising the A-C ratio from 0.21 to 1.52. Under the 50-region setting, the efficiency advantage is even more pronounced, with only 427.8 evaluations required and an A-C ratio of 6.37. On LVIS rare-class detection, overall faithfulness scores are lower due to the long-tail distribution, but the same pattern holds. VPS achieves the highest raw insertion metrics, while Faster-VPS provides comparable results with far fewer model evaluations. For example, Faster-VPS (50) requires just 348.4 evaluations compared to 10100 for VPS, lifting the A-C ratio from 0.18 to 3.76. These results show that Faster-VPS offers an effective trade-off in undetected failure analysis: it

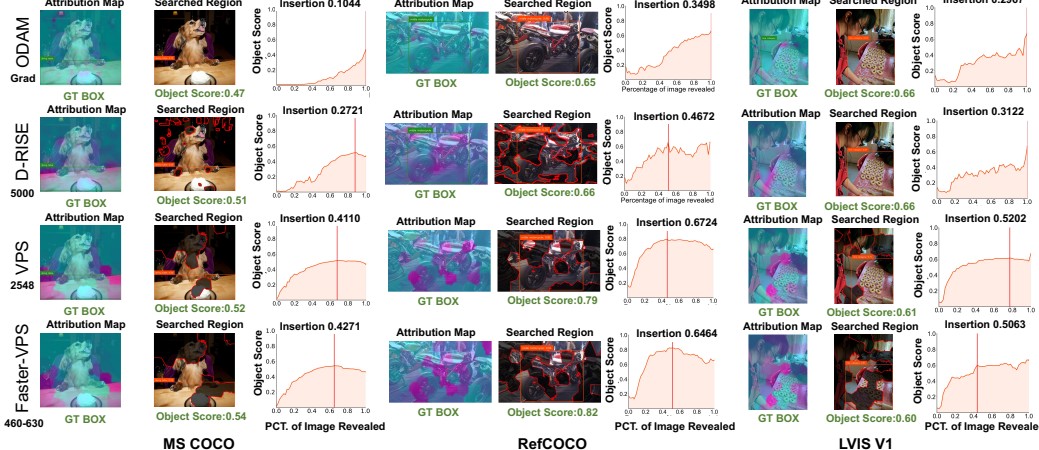

Figure 3: Visualization of correct attribution cases on MS COCO, RefCOCO, and LVIS V1. Compared with ODAM and D-RISE, Faster-VPS produces sharper and more faithful attributions. It matches or even exceeds VPS in insertion-AUC while requiring only ∼20% of its computational cost.

maintains interpretability comparable to VPS while drastically reducing computational cost, enabling practical attribution studies even on large-scale failure cases.

## 4.5 SPEED AND PRECISION CONTROL

An appealing property of our acceleration algorithm is the ability to balance efficiency and accuracy through hyperparameter tuning. By slightly relaxing the speed constraint, Faster-VPS (50) can steadily improve its attribution quality. As shown in Figure 4, the insertion AUC increases monotonically with the number of model forward passes, approaching the performance of the greedy algorithm. Owing to the annealing strategy, our method can even surpass greedy search when fully trading off speed, demonstrating that efficiency and precision can be adaptively controlled.

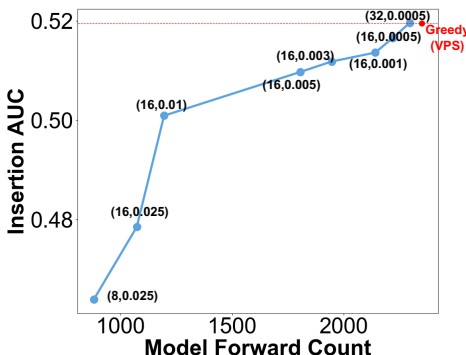

Figure 4: Trade-off between speed and precision.

## 4.6 VISUALIZATION

We further present visualization results for correctly attributed cases, As shown in Figure 3, ODAM produces diffuse heatmaps, while D-RISE generates noisy regions due to perturbation sampling. VPS (50) yields sharp attributions but at a prohibitive computational cost. In contrast, our Faster-VPS (50) achieves nearly the same attribution quality with only about 20% of the overhead, and its annealing strategy often allows the max object score to surpass VPS by better exploring the maximum submodular subset. More visualization results are included in Appendix G.

## 5 CONCLUSION

In this work, we addressed the challenge of efficient attribution for large multimodal foundation models in object detection. Building on the submodular hypothesis and task-specific properties, we proposed the PhaseWin algorithm as a replacement for the original greedy attribution search. Integrated into the Faster-VPS pipeline, our approach achieves up to 95% of state-of-the-art attribution fidelity while requiring only 20% of the computational overhead, establishing the current best practice for efficient interpretation of object-level multimodal models. Beyond object detection, the general applicability of our algorithm to image data suggests promising opportunities for extending this framework to a broader range of multimodal foundation models, which we leave for future research.

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

ACKNOWLEDGMENT OF LLM USAGE

During the preparation of this manuscript, large language models (LLMs) were employed in a limited and auxiliary capacity. Specifically, their usage was restricted to the following three aspects: (1) checking grammar and expression at the sentence level, thereby providing local linguistic refinement; (2) performing global polishing after the draft was completed, ensuring that the overall exposition conforms to idiomatic English usage; and (3) improving the readability of the proof details presented in the appendix.

At no stage were LLMs used for generating research ideas, developing arguments, or modifying the substantive content of this work. Their sole role was to assist in enhancing clarity and effectiveness of communication.

## A  WINDOW SELECTION POLICES

In this section, we first introduce the four algorithms (described in Table 7) we can choose from for the sub-process designed for a window of the phasewin algorithm.

First, the most basic approach is to apply greedy search within the window, which is also the slowest. Our three subsequent designs all use the submodularity assumption to varying degrees to reduce the number of searches within the window. $\pi_{BA}$ uses an adaptive scaling search strategy, $\pi_{T_2}$ considers two elements with the smallest reduction in combined return, and $\pi_{BAF}$ reduces the number of comparisons by maintaining an upper bound list.

Table 7: Window selection policies $\pi(\cdot)$ used within the `WindowSelection` subroutine.

| Policy | Description |
|--------|-------------|
| $\pi_{\mathrm{LG}}$ | **Local-Greedy:** Picks the top candidate if its gain exceeds $\tau_{\mathrm{sel}}$. |
| $\pi_{\mathrm{BA}}$ | **Beta-Adaptive:** Selects all candidates above an adaptive cut-off based on the window's max gain. |
| $\pi_{\mathrm{T2}}$ | **Top-2:** Jointly selects the top two candidates if their gains are high and their relative gap is small. |
| $\pi_{\mathrm{BAF-B}}$ | **Batched Best-Above w/ Forward-checking:** Processes the window in batches, using cached gains to terminate early and reduce evaluations. |

## B  COMPLETE ALGORITHM PROCESS

The algorithm operates in discrete phases. At the start of each phase, it performs a full evaluation on all remaining candidate regions ($\mathcal{R}$) and greedily selects the single best region to anchor the current search state. This ensures consistent progress. Based on the maximum marginal gain ($G_t$) observed in this step, it computes two adaptive thresholds: a selection threshold $\tau_{\mathrm{sel}}$ to identify high-potential candidates and a pruning threshold $\tau_{\mathrm{del}}$ to discard low-utility regions. This adaptive pruning strategy dynamically narrows the search space, focusing computational resources on the most promising regions.

For the initial phases (controlled by a hyperparameter $m_{\mathrm{active}}$), PhaseWin employs a sliding window of size $w$ over the sorted candidate pool $\mathcal{P}_t$. Within this window, a selection policy $\pi(\cdot)$—such as Beta-Adaptive (BA)—is applied to identify a batch of one or more regions for selection. This allows the model to select complementary regions simultaneously, a capability absent in naive greedy search. To further refine the candidate evaluation, a simulated annealing mechanism may defer the entry of lower-scoring regions into the window, allowing more promising candidates to be assessed first. After $m_{\mathrm{active}}$ phases, the algorithm transitions to a simplified greedy selection over the candidate pool to ensure convergence.

A key innovation of PhaseWin is its *dynamic phase supervision*. We monitor the sequence of marginal gains of the selected regions, $\Delta_i = \mathcal{F}(S_i) - \mathcal{F}(S_{i-1})$. If the current gain drops precipitously compared to the previous one (i.e., $\Delta_i < \theta \cdot \Delta_{i-1}$, where $\theta$ is an adaptive supervision coefficient), it signals a potential breakdown of local submodularity. In this event, the algorithm calculates a

probability $p_{\text{exit}}$ to terminate the current phase prematurely. This probabilistic exit prevents the algorithm from wasting evaluations on a sequence of diminishing returns and allows it to restart with a new anchor region. The complete procedure is detailed in Algorithm 1.

## C    EVALUATION METRICS

**Faithfulness.**    To assess how well an attribution map reflects the model's reasoning, we compute the Insertion and Deletion AUC scores, which quantify the change in model output as the most (or least) important superpixels are progressively revealed or removed Petsiuk et al. (2018). We apply these metrics both to classification confidence and to Intersection-over-Union (IoU), thus capturing the attribution's influence on recognition and localization. We further measure the highest confidence score for any predicted box with $\text{IoU} > 0.5$ relative to the target. For failure cases, we evaluate the *Explaining Successful Rate (ESR)*, which measures whether a saliency map can guide the model to a correct detection for initially misclassified or low-confidence predictions.

**Localization Accuracy.**    We use two established metrics: (i) the *Point Game*, which checks whether the most salient pixel lies inside the ground-truth bounding box, and (ii) the *Energy Point Game*, which extends this by considering the energy concentration of saliency around the target Zhang et al. (2018). These metrics are evaluated only on correctly detected samples.

**Efficiency.**    To provide a fair and hardware-agnostic cost measure, we introduce the *Model Evaluation Count (MEC)* as our primary efficiency metric, where one unit corresponds to a single forward pass through the model. The total MEC reflects the algorithm's runtime cost. Additionally, we define the *Accuracy–Cost Ratio (AC-Ratio)* as the primary performance metric (faithfulness score) multiplied by 1000 and divided by the MEC. This ratio is most meaningful when the faithfulness score meets a predefined quality threshold.

## D    IMPLEMENTATION DETAILS

In all experiments, the ground-truth bounding box $b_{\text{target}}$ and its category $c$ are provided as references for generating attributions. Each image is segmented into 100 sparse sub-regions using the SLICO superpixel algorithm, which serve as the interpretable units.

For Faster-VPS, we apply a window size of 16 when selecting from 50 sub-regions and 32 when selecting from 100 sub-regions. Results are averaged over five random seeds, with variance consistently below 2%.

As the scoring function is not strictly monotonic submodular, the stopping criterion is implemented in a ratio-based form:

$$\frac{S_{k-2}}{S_{k-1}} - \frac{S_{k-1}}{S_k} \leq \tau.$$

We use $\tau = 0.025$ for 50 sub-regions and $\tau = 0.01$ for 100 sub-regions. This criterion ensures numerical stability across different settings.

---

**Algorithm 2** Phase-Window (PhaseWin) Search Algorithm

---

1: **Input:** A set of regions $\mathcal{V}$, desired number of regions $k$, scoring function $\mathcal{F}(\cdot, \mathbf{b}_{\text{target}}, c)$.
2: **Hyperparameters:** Window size $w$, active window phases $m_{\text{active}}$, selection ratio $\alpha_{\text{sel}}$, deletion ratio $\beta_{\text{del}}$, supervision coefficients $\{\theta_t\}$.
3: **Output:** An ordered set $S$ of $k$ regions.
4: $S \leftarrow \emptyset; \mathcal{R} \leftarrow \mathcal{V}; t \leftarrow 0; \Delta_{\text{prev}} \leftarrow \infty$;
5: **while** $|\mathcal{S}| < k$ **and** $\mathcal{R} \neq \emptyset$ **do**
6:     $t \leftarrow t + 1$;
7:     *// — Phase Initialization: Anchor Selection —*
8:     $g_r \leftarrow \mathcal{F}(S \cup \{r\}, \mathbf{b}_{\text{target}}, c)$ for all $r \in \mathcal{R}$;
9:     **if** $\max(g_r) \leq 0$ **then break**;
10:     **end if**
11:     $\alpha_{\text{best}} \leftarrow \arg\max\limits_{r \in \mathcal{R}} g_r$;
12:     $S \leftarrow S \cup \{\alpha_{\text{best}}\}; \mathcal{R} \leftarrow \mathcal{R} \setminus \{\alpha_{\text{best}}\}$;
13:     $\Delta_t \leftarrow g_{\alpha_{\text{best}}}$;
14:     *// — Candidate Generation and Pruning —*
15:     Re-evaluate gains $g_r$ for all $r \in \mathcal{R}$; Let $G_t = \max\limits_{r \in \mathcal{R}} g_r$;
16:     $\tau_{\text{sel}} \leftarrow \alpha_{\text{sel}} \cdot G_t; \tau_{\text{del}} \leftarrow \beta_{\text{del}} \cdot G_t$;
17:     $\mathcal{R} \leftarrow \{r \in \mathcal{R} \mid g_r \geq \tau_{\text{del}}\}$;   */* Prune low-gain regions */*
18:     $\mathcal{P}_t \leftarrow \{r \in \mathcal{R} \mid g_r \geq \tau_{\text{sel}}\} \cup \text{RandomSample}(\{r \in \mathcal{R} \mid g_r < \tau_{\text{sel}}\})$;
19:     Sort $\mathcal{P}_t$ in descending order of gain;
20:     *// — Window-Based or Degenerate Greedy Selection —*
21:     **if** $t \leq m_{\text{active}}$ **then**    */* Windowing Mode */*
22:         Initialize window $W$ with the top $w$ elements of $\mathcal{P}_t$;
23:         **while** $|W| > 0$ **and** $|S| < k$ **do**
24:             $A \leftarrow \pi(W, \mathcal{F}, \tau_{\text{sel}})$;    */* Apply selection policy (e.g., BA) */*
25:             **if** $A = \emptyset$ **then break**;
26:             **end if**
27:             **for all** $\alpha \in A$ **do**
28:                 $\Delta_i \leftarrow \mathcal{F}(S \cup \{\alpha\}, \dots) - \mathcal{F}(S, \dots)$;
29:                 **if** $\Delta_i < \theta_t \cdot \Delta_{\text{prev}}$ **then**    */* Dynamic Supervision Check */*
30:                     Calculate exit probability $p_{\text{exit}}(\Delta_i, \Delta_{\text{prev}}, \theta_t)$;
31:                     **if** $\text{rand}() < p_{\text{exit}}$ **then goto** end_phase;
32:                     **end if**
33:                 **end if**
34:                 $S \leftarrow S \cup \{\alpha\}; \Delta_{\text{prev}} \leftarrow \Delta_i$;
35:             **end for**
36:             Update window $W$ by removing selected elements and refilling from $\mathcal{P}_t$;
37:         **end while**
38:     **else**   */* Degenerate Greedy Mode */*
39:         **for all** $\alpha \in \mathcal{P}_t$ **do**
40:             $\Delta_i \leftarrow \mathcal{F}(S \cup \{\alpha\}, \dots) - \mathcal{F}(S, \dots)$;
41:             **if** $\Delta_i < \theta_t \cdot \Delta_{\text{prev}}$ **then**    */* Dynamic Supervision Check */*
42:                 Calculate exit probability $p_{\text{exit}}(\Delta_i, \Delta_{\text{prev}}, \theta_t)$;
43:                 **if** $\text{rand}() < p_{\text{exit}}$ **then break**;
44:                 **end if**
45:             **end if**
46:             $S \leftarrow S \cup \{\alpha\}; \Delta_{\text{prev}} \leftarrow \Delta_i$;
47:             **if** $|S| \geq k$ **then break**;
48:             **end if**
49:         **end for**
50:     **end if**
51:     **end** phase;
52: **end while**
53: **return** $S$;

---

# E   FULL PROOF

In this section, we will introduce the proof of Theorem 3.1. Property 3.1 is a classic result of combinatorial optimization. If you are interested in Property 3.1, you can find the relevant proof in Edmonds (1970); Fujishige (2005).

**Proof of Theorem 3.1.**

*Proof.* If the parameter for AdaptiveThreshold is $(\alpha, \gamma)$ (for select and delete), the parameter for WindowSelection when $|S| = i$ is $\beta_i$ with $\beta_i$ increasing and $\alpha\beta_i \geq \gamma$.

Let $S_{\text{PhaseWin}} = (v_1, v_2, \ldots, v_k)$, $S_0 = \emptyset$ and $S_i = \{v_1, v_2, \ldots, v_i\}$. Let $\rho_u(V) = \mathcal{F}(V \cup \{u\}) - \mathcal{F}(V)$.

For each $1 \leq i \geq k$ such that $v_i$ is an element directly added into $S_{\text{PhaseWin}}$ without going into WindowSelection, let $\mathcal{R}_i$ to be the set of choosable elements before $v_i$ is selected, $\mathcal{D}_i$ to be the set of deleted elements after $v_i$ is selected in PartitionCandidates. Then we have

$$a_i \triangleq \rho_{v_i}(S_{i-1}) = \max_{j \in \mathcal{R}_i} \rho_j(S_{i-1}),$$

$$\mathcal{D}_i \triangleq \{j \in \mathcal{R}_i \mid \rho_j(S_{i-1}) < \gamma a_i\},$$

$$V_i \triangleq \{j \in \mathcal{R}_i \mid \rho_j(S_{i-1}) > \alpha a_i\};$$

$$W_i \triangleq (e_{i,1}, e_{i,2}, \ldots, e_{i,m_i}) \subseteq V_i \text{ is the maximum sequence such that}$$

$$e_{i,j} = v_{i+j} = \text{argmax}\{\rho_e(S_{i+j-1}) \mid e \in S_i \setminus \{e_{i,1}, \ldots, e_{i,j-1}\}\} \text{ and}$$

$$b_{i,j} \triangleq \rho_{e_{i,j}}(S_{i+j-1}) \geq \beta_{i+j} b_{i,0} \geq \alpha\beta_{i+j} a_i \geq \alpha\beta_{i+j} \max_{j \in \mathcal{R}_i} \rho_j(S_{i+j-1}).$$

Thus for any $1 \leq l \leq k$, we have

$$\rho_{v_l}(S_{l-1}) \geq \alpha\beta_l \max_{j \in \mathcal{R}_l} \rho_j(S_{l-1}).$$

Since $\mathcal{F}$ is increasing and submodular, for any $1 \leq l \leq k$ we have

$$\mathcal{F}(S_{\text{OPT}}) \leq \mathcal{F}(S_{l-1}) + \sum_{j \in (T \setminus S_{l-1}) \cap \mathcal{R}_l} \rho_j(S_{l-1}) + \sum_{m=1}^{l-1} \sum_{x \in (T \setminus S_{l-1}) \cap D_m} \rho_j(S_{l-1})$$

$$\leq \mathcal{F}(S_{l-1}) + \frac{k}{\alpha\beta_l} \rho_{v_l}(S_{l-1}) + k\gamma \sum_{m=1}^{l-1} \rho_{v_m}(S_{m-1})$$

$$= \frac{k}{\alpha\beta_l} \mathcal{F}(S_l) - (\frac{k}{\alpha\beta_l} - 1 - k\gamma) \mathcal{F}(S_{l-1}).$$

Let $\lambda_i = \frac{\alpha\beta_i}{k}$ and $\mu_i = \frac{\alpha\beta_i}{k}(\frac{k}{\alpha\beta_i} - 1 - k\gamma)$, then we have

$$\mathcal{F}(S_{\text{PhaseWin}}) \geq \mathcal{F}(S_{\text{OPT}}) \cdot (\lambda_k + \mu_k \lambda_{k-1} + \mu_k \mu_{k-1} \lambda_{k-2} + \cdots + \mu_k \mu_{k-1} \ldots \mu_2 \lambda_1).$$

In particular, if $\beta_i = \beta$ for $i = 1, 2, \ldots, k$, then

$$\mathcal{F}(S_{\text{PhaseWin}}) \geq \frac{\lambda_1 (1 - \mu_1^k)}{1 - \mu_1} \mathcal{F}(S_{\text{OPT}}).$$

If $k, \alpha, \beta$ is big enough and $\gamma$ is small enough, then

$$\mathcal{F}(S_{\text{PhaseWin}}) \geq (1 - \frac{1}{e} - o(1)) \mathcal{F}(S_{\text{OPT}}).$$

$\square$

# F SUBMODULARITY AND SUPERMODULARITY

## F.1 DEFINITIONS

Let $V$ denote a finite ground set of candidate regions and $F : 2^V \to \mathbb{R}$ be a set function that scores any subset $S \subseteq V$.

**Definition F.1** (Submodularity). A set function $F$ is *submodular* if it satisfies the *diminishing returns property*: for all $A \subseteq B \subseteq V$ and $\alpha \in V \setminus B$,

$$F(A \cup \{\alpha\}) - F(A) \ \geq \ F(B \cup \{\alpha\}) - F(B).$$

That is, the marginal gain of adding an element decreases as the context grows.

**Definition F.2** (Supermodularity). A set function $F$ is *supermodular* if it satisfies the *increasing returns property*: for all $A \subseteq B \subseteq V$ and $\alpha \in V \setminus B$,

$$F(A \cup \{\alpha\}) - F(A) \ \leq \ F(B \cup \{\alpha\}) - F(B).$$

That is, the marginal gain of adding an element increases as the context grows.

## F.2 OPTIMIZATION SIGNIFICANCE

Submodularity generalizes the notion of convexity to discrete set functions. Maximizing a monotone submodular function under a cardinality constraint admits a simple greedy algorithm with a $(1-1/e)$-approximation guarantee, which is provably optimal under polynomial-time complexity assumptions. In contrast, supermodular functions exhibit cooperative effects, and their maximization is generally intractable, while their minimization is often easier.

## F.3 AUC CURVE PROPERTIES

In attribution evaluation, we consider the insertion process: progressively adding sub-regions $s_1, s_2, \ldots$ into the image. Let

$$\mathrm{AUC}(k) = \frac{1}{|V|} \sum_{j=1}^{k} F(\{s_1, \ldots, s_j\})$$

denote the cumulative insertion-AUC score up to step $k$.

**Theorem F.1.** *If $F$ is submodular, then the insertion AUC curve $\mathrm{AUC}(k)$ is concave in $k$. If $F$ is supermodular, then $\mathrm{AUC}(k)$ is convex in $k$.*

*Sketch.* For submodular $F$, diminishing returns imply that the marginal gain $F(S \cup \{s\}) - F(S)$ is non-increasing in $|S|$. Thus, the discrete derivative of $\mathrm{AUC}(k)$ decreases with $k$, yielding concavity. Conversely, if $F$ is supermodular, marginal gains increase with $k$, so $\mathrm{AUC}(k)$ is convex. $\qquad\square$

## F.4 IMPLICATIONS FOR DEEP LEARNING

Deep neural networks do not strictly satisfy either submodularity or supermodularity. Instead, their attribution behavior reflects a hybrid of both: some features exhibit redundancy (submodular-like), while others rely on synergy (supermodular-like). From the perspective of distributed computation, submodularity and supermodularity describe not universal properties of the model but rather the modes of feature aggregation. Submodularity corresponds to robust, redundant feature usage, while supermodularity corresponds to cooperative, highly interactive feature combinations. These patterns shed light on how models internally organize basic feature units, rather than providing exact guarantees.

The two models we selected are, respectively, dominated by submodularity and supermodularity. Below, we show the Insertion AUC curves (Figure 5) for Grounding DINO and Florence-2 on the same sample. Their unevenness indicates that Grounding DINO exhibits submodularity most of the time, while Florence-2 is almost universally submodular. Our algorithm achieved acceleration on both models, and the difference in performance is precisely due to the difference between submodularity and supermodularity.

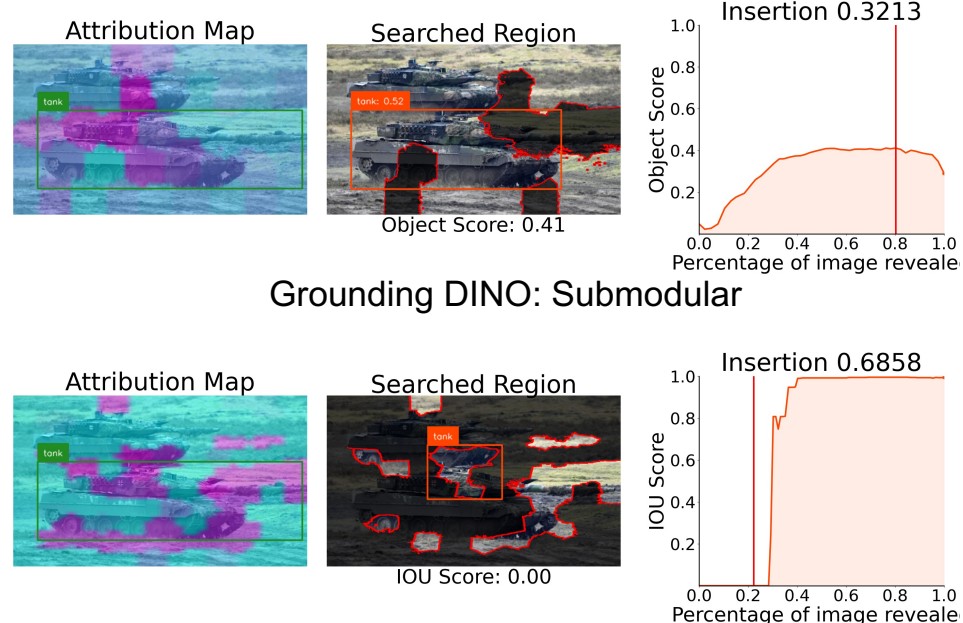

Grounding DINO: Submodular

Florence-2: Supermodular

Figure 5: Insertion AUC under Greedy (VPS). Grounding DINO is almost concave with only a few exceptions, while Florence-2 is completely convex.

## G ADDITIONAL VISUALIZATION RESULTS

In this section, we provide additional qualitative results to further illustrate the visual differences between the original Visual Precision Search (VPS) and our accelerated Faster-VPS. Each figure presents one representative example from different tasks and datasets. For each case, we show side-by-side attribution maps highlighting how both methods localize critical regions that drive the prediction of object-level foundation models. These examples complement the quantitative results in Section 4, demonstrating that Faster-VPS preserves interpretability quality while achieving substantial efficiency gains.

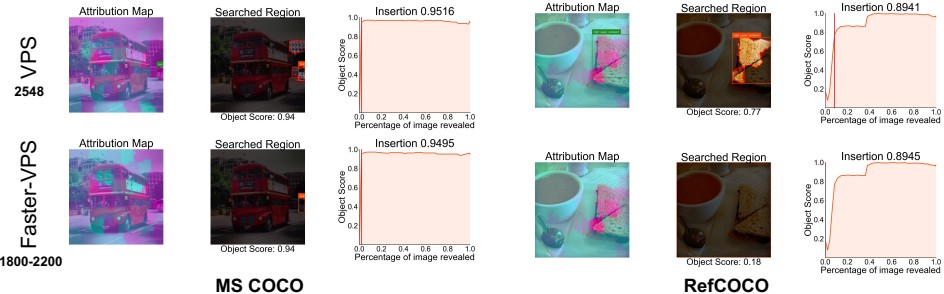

Figure 6: Comparison between VPS and Faster-VPS on Florence-2 for correctly detected samples in MS COCO and RefCOCO. Both methods highlight semantically relevant regions, while Faster-VPS produces equally faithful maps with far fewer evaluations.

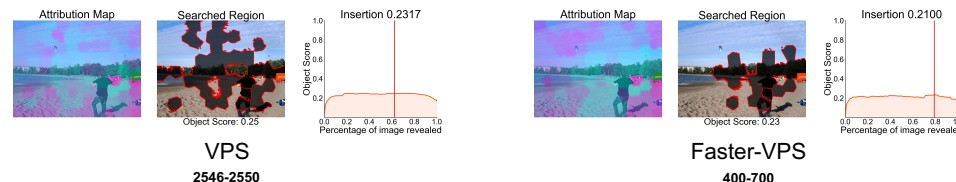

Figure 7: Visualization on Grounding DINO (MS COCO misclassification). VPS and Faster-VPS consistently attribute the incorrect prediction to the same misleading region, confirming that Faster-VPS maintains interpretive fidelity even in failure cases.

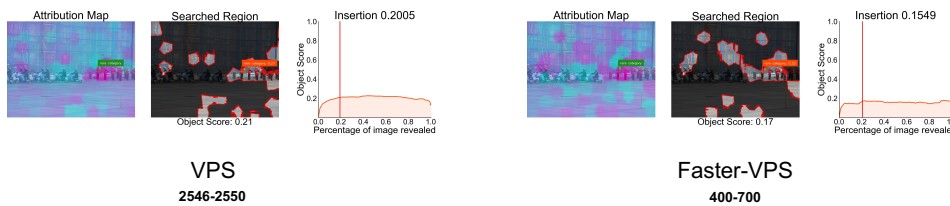

Figure 8: Visualization on Grounding DINO (LVIS misclassification). Both methods reveal the background context responsible for confusion, with Faster-VPS matching the fine-grained localization quality of VPS at lower computational cost.

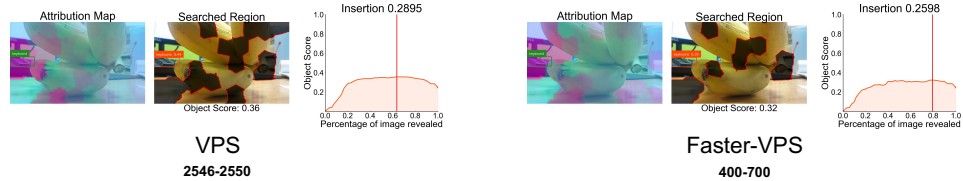

Figure 9: Visualization on Grounding DINO (MS COCO missed detection). VPS and Faster-VPS identify the overlooked object region. Faster-VPS effectively reproduces the trajectory of evidence accumulation with a fraction of the overhead.

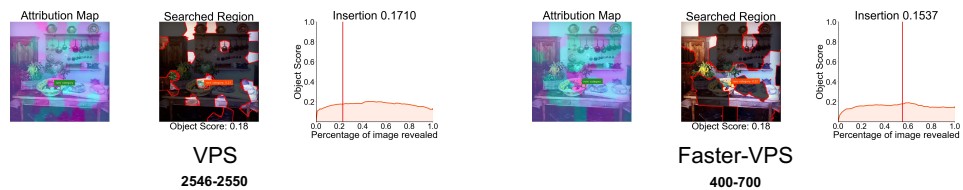

Figure 10: Visualization on Grounding DINO (LVIS missed detection). Faster-VPS successfully recovers the same key evidence regions highlighted by VPS, showing its robustness on challenging zero-shot categories.

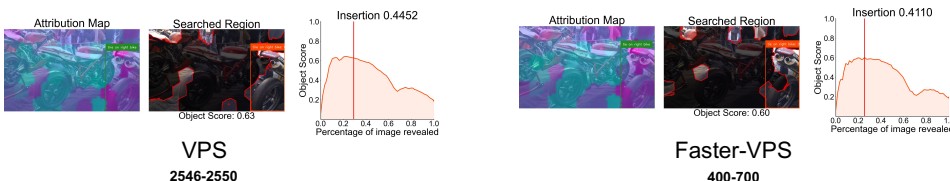

Figure 11: Visualization on Grounding DINO (RefCOCO grounding mistake). Both methods attribute the grounding failure to distractor regions, while Faster-VPS provides nearly identical explanations with significantly fewer model evaluations.

