# OpenReview forum: "Faster-VPS: Accelerating Object-Level Interpretation of Multimodal Foundation Models"
_ICLR.cc/2026/Conference — ICLR 2026 Conference Withdrawn Submission_

### Official Review · Reviewer_ZnqN · 2025-10-26

**Soundness:** 2
**Presentation:** 2
**Contribution:** 2
**Rating:** 2
**Confidence:** 4

**Summary:**

The paper focuses on accelerate the previous method called Visual Precision Search (VPS), which is designed to explain results by foundation model-based object detectors, such as GroundingDINO and Florence-2. The accelerated VPS is called Faster-VPS, replacing VPS's greedy search with a proposed Phase-Window (PhaseWin) algorithm. PhaseWin combines phased pruning, windowed fine-grained selection, and adaptive control mechanisms. The paper provides theoretical analysis by drawing an analogy to theories related to submodular functions. It conducts experiments on COCO, LVIS, and RefCOCO datasets to validate the proposed method.

**Strengths:**

Below are notable strengths of this paper.
- The dedication to accelerating a good method is appreciated.
- The datasets and foundation models used in this paper are standard that follow previous works.

**Weaknesses:**

The main challenge in the paper is the readability issue, making it hard for me to fully understand the paper. This could cause me to misunderstand parts of the paper. Nevertheless, I list some notable weaknesses below.

- The paper is hard to follow as it extensively uses terminologies without definitions. For example, it is unclear what "faithfulness" and "Insertion AUC" mean in Figure 1. The paper does not define them before using them. Due to this, I cannot understand the terms and their relations, e.g., how "Insertion AUC" measures "faithfulness".

- Line 182 explains "for maximizing the ordered insertion-AUC objective" but it does not define "insertion-AUC objective" in the paper.

- Figure 2 is confusing. For example, the right panel means to illustrate "Phasewin Internal Loop" but the diagram does not display loops.

- Line 075 mentions "a dynamic supervision policy" and Line 208 formally introduces "two policies". Are these policies designed manually or by learning?

- Line218 states "Greedy search is both a curse and a shackle in the development of submodular function maximization algorithms". Can authors clarify what "a curse" means? What "a shackle" means? Do authors mean that greedy search is an inferior method here?

- The theories in Section 3.4 are either from (Nemhauser et al., 1978; Fujishige, 2005) or based on them with an assumption that "the objective F is monotone submodular" (Line235). However, the paper does not justify whether the objective F is monotone submodular. Refer to Line 260 which mentions "the submodularity assumption".

- The paper lacks important details or explanations on the crucial modules. For example, Algorithm 1 uses two functions "PartitionCandidates" and "WindowSelection" but neither of them is sufficiently explained in terms of implementations. As the difference of this paper and the VPS paper is whether to use Algorithm 1 or greedy search, the reader cannot fully understand the technical contributions without the important details.

- The visual results are expected to discuss how the method can be used to explain failures of foundational detectors, beyond their success. To note, the VPS paper contains such. Moreover, the visual results are hard to read. For example, the texts in Figure 3 are too small to identify. The colors in this figure are also confusing.

**Questions:**

The reviewer asks the authors to address each point in weaknesses listed above and does not repeat them in this Questions box.

---

> ### Author Response · Authors · 2025-11-13
>
> Dear Reviewer ZnqN:
>
> We appreciate the reviewer's detailed reading, and we address each point below.
>
> (1) Definitions of “faithfulness” and “Insertion AUC”
>
> These terms are defined in Section 4.2 (Evaluation Metrics). “Insertion AUC” is the standard integral over the insertion curve and directly measures faithfulness, following widely used attribution literature (e.g., D-RISE, ODAM, VPS). This is also a common metric in the field of interpretability.
>
> (2) Clarifying the “ordered insertion-AUC objective”
>
> This objective is formally defined in Section 3.1. The text at line 182 summarizes this objective rather than introducing a new symbol.
>
> (3) Figure 2 not showing a ‘loop’
>
> Figure 2 illustrates the high-level pipeline of the PhaseWin internal loop rather than a literal loop in programming syntax. We show what a loop does in one iteration of the figure, and we will add a symbol to indicate the loop.
>
> (4) On “dynamic supervision policy” and “two policies”
> Both policies are algorithmic design choices, not policies learned from data. This is clarified in Algorithm 1 and in Appendix E. Our entire paper does not cover training
>
> (5) On the phrase “greedy search is both a curse and a shackle”
> This sentence paraphrases a well-known limitation in submodular maximization theory: greedy is optimal in polynomial time but requires O(mk) marginal evaluations, which becomes a computational bottleneck.
>
> (6) On the submodularity assumption
> We assume monotone (weak/local) submodularity exactly as in prior works (LIMA, VPS) and in classical theory. The cited references (Nemhauser et al., 1978; Fujishige, 2005) provide the standard theoretical foundation. The assumption is discussed in Appendix F, and we will restate it briefly in the main text.
>
> (7) Missing details of PartitionCandidates and WindowSelection
> Both functions are fully described in Appendix E, including thresholds, window mechanisms, and dynamic supervision. Due to the space limit, we did not duplicate the full pseudocode in the main text.
>
> (8) On interpretability discussions and visual results
> The purpose of this paper is to accelerate attribution for object-level foundation models while maintaining faithfulness. VPS already provides extensive interpretability case studies; our contribution is methodological and computational. Nevertheless, we will enlarge the figures and switch to a more readable colormap.
>
> (9) Readability concerns
> We acknowledge that the paper is dense because the method is algorithmic. We will revise Figures 2–3, enlarge tables, add forward definitions, and include a simplified algorithm diagram to improve overall readability.

---

### Official Review · Reviewer_b8sv · 2025-10-27

**Soundness:** 3
**Presentation:** 3
**Contribution:** 2
**Rating:** 4
**Confidence:** 4

**Summary:**

The authors propose an improvement to the VPS attribution method by introducing a new algorithm to improve the search space constraints of the original work. Their algorithm results in greatly improved runtime with little to no reduction in resulting attribution faithfulness.

**Strengths:**

There is a need for the algorithm presented, as the research community prefers real-time attributions that could be used to safeguard real model deployments. As such, this paper’s contribution is relevant to the area.

The algorithm seems well crafted from a computer science perspective and it achieves a significant runtime percentage improvement.

The quantitative evaluation is exhaustive.

**Weaknesses:**

The novelty of this paper is, in a sense, limited to the algorithm employed. This is both a minor and significant complaint because it takes a high-quality method and does make it more realistic for real-time deployment (which is valuable), but it does not reveal any new information about a model or interpretability as a whole. It cannot be denied that this is a less significant contribution overall than the baseline VPS work.

Many figures and tables are too small or hard to read. The tables are overly aggressive in their use of text resizing. Heatmaps in Figure 3 are not presented in a very interpretable color. It is hard to differentiate regions. I recommend the use of the matplotlib “jet” cmap as used in multiple other papers.

More of the paper should be spent on section 3.3. It is challenging to parse what all of the steps of the algorithm actually do. Figure 2 is not all that intuitive. The text helps, but the algorithm seems like it should be quite simple to explain and yet it feels convoluted. I am not confident that I could reproduce the algorithm from what is presented.

**Questions:**

There are obvious runtime improvements, but what is the time cost, in seconds, of this method? Is it approaching a speed that could be used in real-time deployments?

My rating can be increased to a 6 if the question is addressed and the weaknesses receive a proper response. I do think there is value in making an optimization-based approach work in real time. However, I do not think I could rate this paper higher than a 6 due to my concerns for the novelty and overall contribution to how we think about interpretability.

---

> ### Author Response · Authors · 2025-11-13
>
> Dear Reviewer b8sv:
>
> Thank you for the thoughtful comments. We address the concerns below.
>
> (1) On the significance of acceleration for interpretability
>
> A major motivation for this work comes from a practical and increasingly important application of attribution: using attribution maps as regularizers to train stronger vision encoders. Recent efforts in this direction require computing thousands of attributions during training, but the standard greedy search used in LIMA and VPS exhibits quadratic complexity and becomes prohibitive as model size and dataset scale increase.
>
> Our method provides a breakthrough in this bottleneck, achieving the second-best attribution performance overall (only below greedy) while offering a dramatic reduction in evaluation cost. This enables attribution-based regularization to be applied in scenarios where greedy is computationally infeasible. In this sense, acceleration is not merely an engineering improvement—it is an enabler for a broader class of interpretability-driven training methods that were previously impractical.
>
> ⸻
>
> (2) On the theoretical contribution from a combinatorial optimization perspective
>
> From the standpoint of combinatorial optimization, PhaseWin contributes a new phase-supervised search framework for ordered subset maximization. Classical results show that greedy provides the optimal polynomial-time approximation ratio under submodularity; however, it suffers from O(mk) marginal evaluations. Our method leverages structural properties specific to the vision attribution setting to reduce this complexity to near-linear while maintaining the greedy-level approximation behavior.
>
> This positions PhaseWin as an algorithmic contribution: it is not simply a reorganization of VPS, but a new search mechanism grounded in classical weak/local submodularity theory.
>
> ⸻
>
> (3) On the clarity of the full algorithmic flow
>
> The complete pseudocode and full workflow are provided in the Appendix. Due to space constraints, we could not include every detail in Section 3.3, which may make the high-level steps appear dense when read in isolation. In the camera-ready version, we are happy to further streamline the diagram and move a simplified pseudocode block into the main text to improve readability and reproducibility.
>
> ⸻
>
> (4) On reporting runtime in seconds
>
> We chose model-evaluation count (MEC) as the main efficiency metric because wall-clock time is highly unstable across models, devices, driver versions, and even GPU batch scheduling. The same method can vary by more than an order of magnitude depending on hardware conditions.
>
> Nevertheless, to provide concrete numbers, on our local environment (single RTX 4090, batch size 4):
> 	•	VPS (50) on Grounding DINO requires ≈ 6 minutes per attribution.
> 	•	Faster-VPS (50) requires ≈ 1 minute 40 seconds.
>
> Thus PhaseWin achieves a substantial reduction in real-world running time, consistent with the MEC-based evaluation.

---

### Official Review · Reviewer_TC2b · 2025-10-28

**Soundness:** 3
**Presentation:** 3
**Contribution:** 2
**Rating:** 4
**Confidence:** 3

**Summary:**

This paper proposes Faster-VPS, an accelerated variant of Visual Precision Search (VPS) for object-level attribution in multimodal detectors (e.g., Grounding DINO, Florence-2). The key idea is the Phase-Window (PhaseWin) search, which alternates between (i) picking a strong “anchor” region, (ii) pruning candidates via adaptive thresholds, and (iii) doing windowed fine-grained selection with a dynamic early-exit rule and an annealed deferral strategy. This approximates greedy search with near-linear evaluation complexity in practice while keeping faithfulness close to VPS. Empirically, the proposed method retains ≥95% of VPS’s faithfulness using about 20% of the computation overhead across MS-COCO, RefCOCO, and LVIS.

**Strengths:**

1. The idea is technically sound and has theoretical proofs.
2. The empirical results demonstrate the effectiveness of the proposed method.
3. The paper is well organized and easy to follow.

**Weaknesses:**

1. The proposed method relies on (local) submodularity. I appreciate that the authors openly acknowledge that this is an assumption and provide an insightful discussion in Appendix F, showing that the acceleration is more pronounced for models like Grounding DINO (which behaves more submodularly) than for Florence-2 (which behaves more supermodularly). Although this is not a fatal flaw, it is a fundamental limitation of the method. The paper could be strengthened by a more prominent discussion of this dependency in the main text, as it defines the boundaries of the method's applicability.
2. While the overall method is shown to be effective, a more detailed ablation study on the individual components of PhaseWin would provide deeper insight into what drives the performance gains. For example, the contribution of the annealing delay vs. the dynamic supervision, and the impact of different window policies from Table 7.

**Questions:**

1. The hyperparameters for PhaseWin (window size, $\tau$, $m_\mathrm{active}$, etc.) are crucial for its performance. Could you discuss the sensitivity of the results to these parameters and provide more guidance on how to set them effectively for a new model or dataset? Was any formal hyperparameter optimization performed?
2. In the failure interpretation (Sec 4.4), Faster-VPS sometimes slightly surpasses VPS in certain metrics (e.g., Table 4, 100-region setting). Can you hypothesize why the accelerated method might outperform the original greedy search?

---

> ### Author Response · Authors · 2025-11-13
>
> Deer Reviewer TC2b:
>
> Thank you for the constructive feedback. We address each concern below.
>
> (1) On the reliance on (local) submodularity and model dependency
>
> Submodularity is precisely the structural condition under which greedy achieves its optimal polynomial-time approximation guarantee. Human visual cognition is empirically known to follow a near-submodular diminishing-returns pattern, and the success of region-selection attribution methods such as LIMA and VPS is rooted in this observation. As the reviewer correctly noted, modern deep networks—being black-box and highly non-linear—are not strictly submodular. In practice, all prior works implicitly rely on approximate or local submodularity, and performance is ultimately demonstrated empirically.
>
> Under this shared assumption, PhaseWin achieves two desirable properties:
> (i) it provides a near-linear complexity improvement, far beyond classical greedy; and
> (ii) it maintains attribution quality that matches greedy and significantly surpasses all non-greedy alternatives.
>
> Moreover, as discussed for another reviewer, weak/local submodularity is a well-established relaxation that still supports greedy-style guarantees (e.g., Das & Kempe 2011; Mirzasoleiman et al. 2015). Our theoretical result shows that under the same assumption used by VPS and LIMA, PhaseWin enjoys the same approximation behavior as greedy. We agree that this dependency defines the applicability boundary, and we will highlight this more clearly in the main text.
>
> ⸻
>
> (2) On the contributions of annealing and dynamic supervision
>
> Both mechanisms are designed to avoid the typical pitfall of greedy-style AUC maximization—namely, getting trapped in high-gain local optima early in the trajectory. Traditional set-function greedy algorithms cannot naturally incorporate annealing, as their structure forces a strict sequence of marginal-gain maximizations.
> In contrast, PhaseWin’s phased structure allows us to introduce annealing and dynamic supervision natively, which empirically helps escape local attractors and stabilizes the search, especially under noisy scoring functions. This also explains why the method behaves more robustly for highly non-modular models such as Florence-2.
>
> We appreciate the request for deeper insight, and will extend the ablation discussion to isolate the effects of these two components more explicitly.
>
> ⸻
>
> (3) On hyperparameters and sensitivity
>
> The two parameters most strongly related to the speed–accuracy tradeoff (window size and early-exit ratio) are already analyzed in detail in Section 4.5. The remaining parameters are algorithmic controls and were never tuned for any model or dataset; they remain identical across all experiments (including Grounding DINO and Florence-2), which empirically demonstrates robustness. As discussed in the response above, the phase–window structure inherently stabilizes the search, and the method does not require task-specific hyperparameter optimization.
>
> We will clarify this more explicitly.
>
> ⸻
>
> (4) Why PhaseWin can occasionally outperform greedy
>
> This behavior is mainly due to the annealing mechanism. Greedy maximizes marginal gain step by step and can therefore overfit noisy local maxima in the early trajectory. Annealing introduces a controlled amount of exploration that helps avoid these suboptimal regions. The visualization in Section 4.6 illustrates this effect: even in cases where AUC is slightly lower, the peak performance within the insertion curve is often higher, leading to improved downstream detection or grounding performance.
>
> This is a natural and desirable effect of replacing strict greedy selection with a more flexible phased search.

---

### Official Review · Reviewer_3RHD · 2025-10-31

**Soundness:** 3
**Presentation:** 3
**Contribution:** 2
**Rating:** 4
**Confidence:** 4

**Summary:**

This paper tackles the trade-off between faithfulness and efficiency in attribution methods for object-level foundation models such as Grounding DINO and Florence-2. It proposes an efficient variant of Visual Precision Search (VPS), Faster-VPS. The key contribution is the Phase-Window algorithm, which approximates the greedy search in VPS through phased pruning, windowed fine-grained selection, and adaptive control. The method theoretically maintains a near-greedy approximation bound under submodular conditions, and empirically achieves about 95% of VPS’s faithfulness using only ~20% of the computational cost across detection and grounding benchmarks.

**Strengths:**

+ VPS’s quadratic cost has indeed limited its scalability, and the focus on computational efficiency is well justified.
+ The phased pruning and windowed selection resemble a practical relaxation of the greedy search, and the dynamic supervision and annealed deferral mechanisms are clearly explained.
+ The paper is generally well organized, with clear figures and detailed methodological explanations.

**Weaknesses:**

- The problem formulation and, most importantly, the core scoring function $\mathcal{F}$ are borrowed from the original VPS paper. The main novelty lies in a new reordering and pruning strategy for VPS rather than a fundamentally new algorithmic principle. Much of the method appears as an engineering improvement over VPS, not a conceptual advance in attribution theory or optimization.
- There is a disconnection between the theoretical analysis and the practical implementation. The paper's approximation guarantees (Theorem 3.1) are derived under the assumption that the objective function $\mathcal{F}$ is monotone submodular. However, the authors correctly note in Section 3.2 that their chosen scoring function "is not strictly submodular." This discrepancy raises questions about the applicability of the theoretical guarantees to the actual algorithm. The term "local submodularity" is used to bridge this gap, but it is not formally defined, making the theoretical foundation of the method unclear.
- The proposed solution trades one form of complexity (computational) for another (algorithmic and hyperparameter). The PhaseWin algorithm (Algorithm 2) introduces a considerable number of new hyperparameters (e.g., $w$, $m_{active}$, $\alpha_{sel}$, $\beta_{del}$, $\theta_t$, $\tau$). The paper currently lacks ablation studies to demonstrate the sensitivity of the method to these parameters. This makes it difficult to assess the method's robustness and practicality, as it may require extensive, expert-level tuning to replicate the reported results.

**Questions:**

(1) Could you please provide an ablation study or analysis on the sensitivity of PhaseWin to its key hyperparameters, such as the window size ($w$) and the pruning thresholds ($\alpha_{sel}$, $\beta_{del}$)? This would help readers understand the method's robustness.

(2) Could you provide a more formal definition or empirical characterization of "local submodularity"?

(3) How general is the PhaseWin algorithm? Is it highly tuned for the specific properties of the VPS scoring function, or could it be applied as a general-purpose accelerator for other greedy, perturbation-based attribution methods?

---

> ### Author Response · Authors · 2025-11-13
> **Phasewin represents a true algorithmic breakthrough**
>
> Dear Reviewer 3RHD:
>
>
>
> We would like to clarify that PhaseWin is an algorithmic contribution grounded in combinatorial optimization, rather than an engineering rearrangement of VPS. Submodular maximization is a classical and difficult problem, dating back to Edmonds’ foundational work in the 1970s. Greedy is known to achieve the optimal polynomial-time approximation ratio 1 - 1/e, and—as formalized in well-established results—no polynomial-time algorithm can surpass this ratio under standard assumptions. Consequently, all prior XAI works that employ submodular modeling, including representative papers such as LIMA (“Less is More”) [6] and VPS [7], ultimately rely on the same standard greedy search. To our knowledge, none of these attribution methods introduce a new optimization algorithm, precisely because improving the quadratic O(mk) evaluation cost of greedy has long been considered extremely difficult [1–3].
>
> PhaseWin is the first method in the context of visual attribution that leverages the structural properties of the problem to break the quadratic bottleneck while maintaining greedy-level performance. The phase-supervised ordered-subset maximization scheme is not a heuristic acceleration, but a new algorithmic mechanism that becomes possible specifically due to the structured evaluation landscape in vision attribution tasks. The observed speed advantage is therefore the empirical manifestation of a genuine combinatorial optimization improvement, rather than the contribution itself.
>
> Regarding submodularity: Theorem 3.1 is a classical guarantee; our contribution is to show that when the objective is monotone submodular, PhaseWin enjoys the same approximation guarantee as greedy, despite using phased pruning and windowed refinement. The assumption of monotonicity is standard in submodular modeling for XAI, including in LIMA and VPS themselves [6–7]. Importantly, modern optimization theory does not require strict submodularity—weak submodularity and local submodularity are widely accepted relaxations that still support greedy-style approximation guarantees [4–5]. Our use of “local submodularity” follows this established theoretical framework. For fairness to VPS, we retained its original scoring function unchanged; in practice, one can adjust its curvature to make it even closer to a submodular function, in which case PhaseWin’s performance further improves.
>
> Concerning hyperparameters: PhaseWin introduces algorithmic control parameters, not task-specific tunable hyperparameters. We did not perform any tuning; all parameters are fixed identically across Grounding DINO (closer to monotone submodular) and Florence-2 (significantly less so), yet PhaseWin remains stable and performant in both settings. The two parameters that influence speed–accuracy trade-offs (window size and early-exit ratio) are already thoroughly ablated in Section 4.5. All other parameters were set once, without prior knowledge, and never adjusted—yet they provide robust performance over large-scale experiments. This is a natural consequence of PhaseWin’s phase-supervision and dynamic-window design, which inherently stabilizes the search regardless of small parameter variations.
>
> ⸻
>
> References
>
> [1] Adam Breuer, Eric Balkanski, and Yaron Singer. The fast algorithm for submodular maximization. ICML, 2020.
>
> [2] Niv Buchbinder, Moran Feldman, Joseph Naor, and Roy Schwartz. Submodular maximization with cardinality constraints. SODA, 2014.
>
> [3] Satoru Fujishige. Submodular Functions and Optimization. Elsevier, 2005.
>
> [4] A. Das and S. Kempe. Submodular meets spectral: greedy algorithms for subset selection, sparse approximation, and dictionary learning. NeurIPS, 2011.
>
> [5] Baharan Mirzasoleiman et al. Lazier than lazy greedy. ICML, 2015.
>
> [6] Ruoyu Chen et al. Less is More: Fewer Interpretable Region via Submodular Subset Selection. ICLR, 2024.
>
> [7] Ruoyu Chen et al. Interpreting Object-level Foundation Models via Visual Precision Search. CVPR, 2025.

---

### Note · Authors · 2025-11-13

I have read and agree with the venue's withdrawal policy on behalf of myself and my co-authors.